# Direct-from-specimen microbial growth inhibition spectrums under antibiotic exposure and comparison to conventional antimicrobial susceptibility testing

Jade Chen[1], Su Su Soe San[1], Amelia Kung[1], Michael Tomasek[1], Dakai Liu[2], William Rodgers[2,3], Vincent Gau[1]*

1 GeneFluidics, Los Angeles, California, United States of America, 2 Department of Pathology and Clinical Laboratories, New York-Presbyterian Queens, Flushing, New York, United States of America, 3 Department of Pathology and Laboratory Medicine, Weill Cornell Medical College, New York City, New York, United States of America

* vgau@genefluidics.com

**Data Availability Statement:** All relevant data are within the paper and its Supporting Information files.

## Abstract

Increasing global travel and changes in the environment may escalate the frequency of contact with a natural host carrying an infection and, therefore, increase our chances of encountering microorganisms previously unknown to humans. During an emergency, the etiology of infection may be unknown at the time of patient treatment. The existing local or global Antimicrobial Stewardship Programs may not be fully prepared for emerging/re-emerging infectious disease outbreaks, especially if they are caused by an unknown organism, engineered bioterrorist attack, or rapidly evolving superbug. We demonstrate an antimicrobial efficacy profiling method that can be performed in hours directly from clinical urine specimens. The antimicrobial potency was determined by the level of microbial growth inhibition and compared to conventional antimicrobial susceptibility testing results. The oligonucleotide probe pairs on the sensors were designed to target Gram-negative bacteria, specifically *Enterobacterales* and *Pseudomonas aeruginosa*. A pilot study of 10 remnant clinical specimens from the Clinical Laboratory Improvement Amendments-certified labs of New York-Presbyterian Queens was conducted, and only one sample was not detected by the probes. The remaining nine samples agreed with reference AST methods (Vitek and broth microdilution), resulting in 100% categorical agreement. In a separate feasibility study, we evaluated a dual-kinetic response approach, in which we inoculated two antibiotic stripwells containing the same antimicrobial concentrations with clinical specimens at the original concentration (1x) and at a 10-fold dilution (0.1x) to cover a broader range of microbiological responses. The combined categorical susceptibility reporting of 12 contrived urine specimens was 100% for ciprofloxacin, gentamicin, and meropenem over a range of microbial loads from $10^5$ to $10^8$ CFU/mL.

**Funding:** VG received Award Number RO1AI117059 from the National Institute of Allergy and Infectious Diseases of the National Institutes of Health. https://www.niaid.nih.gov/ VG received Award Number R44HD084033 from the Eunice Kennedy Shriver National Institute of Child Health & Human Development of the National Institutes of Health. https://www.nichd.nih.gov/ Jade Chen, Su Su Soe San, Amelia Kung, Michael Tomasek, and Vincent Gau are employed by GeneFluidics. As a non-academic, commercial company, the employer and funder provided support in the form of salaries for authors [JC, SS, AK, MT, VG], but did not have any additional role in the study design, data collection and analysis, decision to publish, or preparation of the manuscript. The specific roles of these authors are articulated in the 'author contributions' section.

**Competing interests:** As authors of this study [JC, SS, AK, MT, VG], we declare that we are employed by GeneFluidics, a nonacademic, commercial company. This does not alter our adherence to PLOS ONE policies on sharing data and materials. NYPQ has received a contract under a Master Collaboration Agreement from GeneFluidics. The following authors are employed by NYPQ: Dakai Liu, William Rodgers.

# Introduction

Direct-from-specimen microbial growth inhibition assessment can assist in emergency preparedness and pre-hospital interventions by providing timely patient-specific antimicrobial efficacy profiling information. The use of empirical therapy proves that the methods currently used are inadequate when it comes to informing initial treatment decisions in a timely manner [1]. Phenotypic antimicrobial efficacy profiling, in which clinical specimens are directly exposed to different antibiotic conditions, could provide critical information for the prescription of antibiotics in hours. The results of a phenotypic antimicrobial efficacy profile test, taken in conjunction with local antibiogram data, could guide the course of therapy to improve patient outcomes and slow the spread of antimicrobial resistance. We present a molecular test based on the transcriptional responses of causative bacteria to antibiotic exposure that can be performed directly from urine specimens. Quantification of group-specific or species-specific 16S rRNA growth sequences was used to provide rapid antimicrobial efficacy profiling results. Categorical agreement was assessed with reference AST methods according to CLSI guidelines.

Even though antibiotics do not directly affect the SARS-CoV-2 respiratory virus responsible for the COVID-19 pandemic, physicians are administering many more antibiotics than normal when treating COVID-19 patients [2]. The appearing surge in antibiotic use is reflected in the higher percentages of COVID-19 patients with severe conditions and pediatric patients (85% in a multicenter pediatric COVID-19 study [3]) receiving antibiotic therapies. The World Health Organization warned that the use of antibiotic therapy may lead to higher bacterial resistance rates and increase the burden of the pandemic [4]. A recent study by Zhou et al. [5] found that 15% of 191 hospitalized COVID-19 patients, as well as 50% of the 54 non-survivors, acquired bacterial infections. Therefore, a shorter time to rule out certain antibiotic options by detecting microbial growth under such conditions may provide physicians with valuable information before the availability of conventional AST results.

Generating curves to illustrate the microbial growth inhibition response to antibiotic exposure conditions across a range of microbial loads may provide a dynamic method for estimating antimicrobial efficacy that is much more rapid than the endpoint minimum inhibitory concentration (MIC) method used in conventional AST. Here, we present a method to quantify the 16S rRNA content of viable target pathogens in unprocessed specimens, such as urine, following exposure to various antibiotic concentrations *in vitro* (Fig 1). This method allows for interpretation of the antimicrobial effect by analyzing the differential microbial responses at two inoculum dilutions. The hypothesis is that the growth inhibition concentration (GIC) is the lowest antimicrobial concentration necessary to inhibit the growth of target strains in a given sample after adjusting for pathogen concentration effects. The combined GIC in a polymicrobial sample is not evaluated in this pilot study. We compare the GIC reported from this direct-from-specimen antimicrobial efficacy profiling method to the MIC and susceptibility reported from CLSI reference methods to assess the categorical agreement, after which we establish a correlation between the microbiological susceptibility (i.e., MIC) and antimicrobial efficacy (i.e., GIC).

## Electrochemical-based molecular quantification of RNA transcription for streamlined ID and phenotypic AST

Prior to the presented study, we developed a PCR-less RNA quantification method that performs enzymatic signal amplification with a proprietary electrochemical sensor array. We applied this quantification method to a streamlined pathogen identification and AST using species-specific probe pairs, then validated and published studies with our clinical

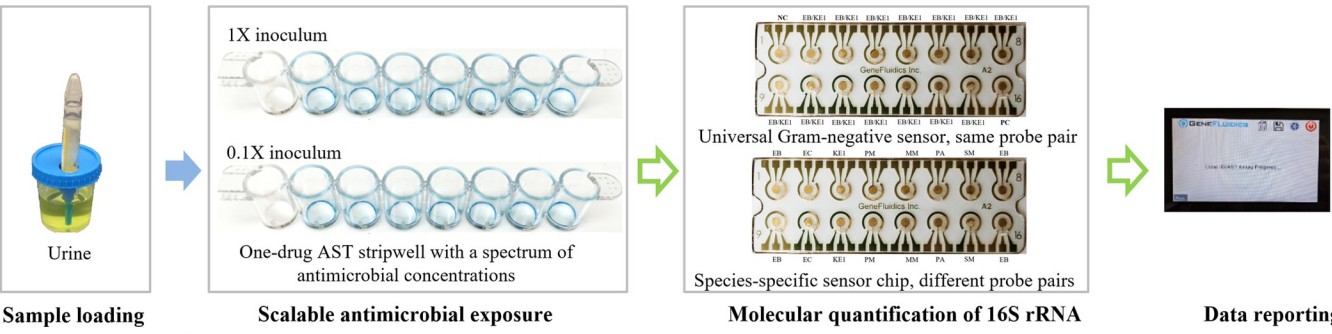

**Fig 1. Graphical abstract of presented direct-from-specimen antimicrobial efficacy profiling method.** Unprocessed urine is inoculated into two antibiotic stripwells at the original concentration and 10-fold dilution. After antibiotic exposure, viable 16S rRNA is quantified using an electrochemical sensor assay and reported as categorical susceptibility. The presented method is able to be fully automated.

collaborators using contrived and remnant clinical specimens [6–36]. The detection strategy of our universal electrochemical sensors is based on sandwich hybridization of capture and detector oligonucleotide probes which target 16S rRNA as described in S1 Protocol. The capture probe is anchored to the gold sensor surface, and the detector probe is linked to horseradish peroxidase (HRP). When a substrate such as 3,3',5,5'-tetramethylbenzidine (TMB) is added to an electrode with capture-target-detector complexes bound to its surface, the substrate is oxidized by HRP and reduced by the bias potential applied onto the working electrode. Oligonucleotide probe sequences for both capture and detector probes are detailed in Fig 2 of S1 Protocol. This redox cycle results in the shuttling of electrons by the substrate from the electrode to the HRP, producing enzymatic signal amplification of current flow in the electrode. The concentration of the RNA target captured on the sensor surface can be quantified by the reduction current measured through the redox reaction between the TMB and HRP with a multi-channel potentiostat built into our system as demonstrated in Fig 4 of S1 Protocol. Quantifying changes in RNA transcription appears to be a more suitable approach in the case of timely reporting due to its rapid changes upon exposure to antibiotics [37, 38]. Measuring the RNA response of pathogens to antibiotic exposure directly in clinical specimens would provide a rapid susceptibility assessment that can be performed in clinical settings.

## Material and methods

### Bacterial strains and antibiotic stripwells

Strains were obtained from various sources including the CDC AR Bank and New York-Presbyterian Queens (NYPQ). The number of strains of each species is listed in S1 Table. All clinical isolates were obtained anonymously from remnant patient samples collected for routine culture and were de-identified prior to testing under the approved NYP/Queens Institutional Review Board and joint master agreement. We aimed to test an even distribution of species with MIC values on or near the susceptible and resistant breakpoints of each antibiotic. We included three representative antibiotics of three different classes (fluoroquinolones, aminoglycosides, and carbapenems): ciprofloxacin (CIP; Cayman Chemical Company, Ann Arbor, MI), gentamicin (GEN; Sigma-Aldrich, St. Louis, MO), and meropenem (MEM; Cayman Chemical Company). CDC AR Bank isolates were used to include representative bacteria susceptibility profiles that were not covered by those from NYPQ. CDC AR Bank isolates were stored as glycerol stocks at -80°C and were grown from these stocks at 35°C on tryptic soy

agar plates with 5% sheep's blood (Hardy Diagnostics) for 18–24 hours before testing. Suspensions of each isolate to be used for contriving urine samples were prepared using cation-adjusted Mueller-Hinton II (MH) broth (Teknova; Hollister, CA) and a Grant DEN-1B densitometer (Grant Instruments, Cambridge, UK). Negative urine specimens to be used for contrived samples were stored in Falcon tubes at 4˚C. Clinical urine samples from NYPQ were stored in BD 364954 Vacutainer Plus C&S tubes containing boric acid at 4˚C prior to overnight shipment for testing. Consumables consisted of stripwells with dried antibiotics, electrochemical-based sensor chips functionalized with oligonucleotide probe pairs complementary to *Enterobacterales* and *Pseudomonas aeruginosa* for RNA quantification (probe sequences in Fig 2A of S1 Protocol), and a reagent kit for lysing and viability culture. Stripwells were prepared as previously described by drying antibiotics in DI water with 0.1% Tween onto EIA/RIA 8-well strips (Corning, Corning, NY) at the following concentrations: CIP 0.0625, 0.125, 0.25, 0.5, 1, 2, 4 μg/mL; GEN 1, 2, 4, 8, 16, 32, 64 μg/mL; MEM 0.5, 1, 2, 4, 8, 16, 32 μg/mL [39]. The first well of each stripwell was left without antibiotic to be used as a growth control (GC) during the assay. Electrochemical sensor chips were produced in-house by deposition of gold onto a plastic substrate and functionalized with probes as previously described [35].

## Specimen collection and matrix removal

Urine samples were spun down to remove the majority of matrix components in the supernatant. Specifically, urine samples with a 4-mL starting volume were spun in a centrifuge at 5,000 RPM for 5 minutes, after which supernatant was removed and replaced with 4 mL of cation-adjusted MH broth to create the 1x inoculum. A ten-fold dilution of this sample was prepared by adding 100 μL of this sample to 900 μL of MH broth, resulting in the 0.1x inoculum.

## Electrochemical-based microbial growth quantification

The direct-from-specimen antimicrobial efficacy profiling approach presented in this study aims to demonstrate a significant correlation to conventional AST results. The electrochemical-based biosensor measures the reduction current from cyclic enzymatic amplification of an HRP label with TMB and $H_2O_2$. The resulting reduction current signal can be estimated with the Cottrell equation (Equation 1 of S1 Protocol) [40]. Signal levels (in nanoamperes) from each microbial exposure well were normalized to that of the GC well (no antibiotics) to form GC ratios. These ratios were then plotted against the spectrum of antimicrobial concentrations tested for statistical analysis. Two antibiotic stripwells containing the same range of seven antibiotic concentrations separated by twofold dilutions, as well as one GC, were used for each specimen at 1x (undiluted pellet) and 0.1x (diluted pellet) concentrations to generate two microbial response curves. Each dual-response curve signature was generated by overlaying the two GC ratio curves over the antibiotic range, establishing a signature library that corresponded to each antimicrobial efficacy and microbial susceptibility combination. Changes in the response signature and inflection point in the GC curve were analyzed by three algorithms to match a categorical classification (susceptible, intermediate, or resistant).

## Antibiotic exposure stripwell inoculation and molecular quantification

One hundred microliters of reconstituted specimen pellets at 1x and 0.1x concentrations were inoculated into each well of their corresponding antibiotic stripwell. All stripwells were incubated at 35˚C for the exposure time indicated in each study. Thirty-six microliters of 1M NaOH were then added to each well to lyse target gram-negative pathogens after antibiotic exposure, followed by a 3-minute incubation at room temperature. Twenty-four microliters of 1M HCl were then added to each well to neutralize the pH of the lysed sample, or lysate, and

prevent the degradation of free RNA. Ten microliters of the lysate from each well were pipetted onto its corresponding sensors on two electrochemical sensor chips for a total of 4 sensors per well. No sample was delivered to the negative control sensor. All chips were incubated for 30 minutes at 43˚C, and the RNA content was quantified using the method described above and in S1 Protocol to obtain the microbial growth response.

## Clinical feasibility validation with blind clinical specimens

For the blind testing study, we used remnant clinical specimens collected at NYPQ under the current IRB. These urine specimens were prospectively collected for urine culture as part of routine care. All samples shipped overnight to GeneFluidics for testing were confirmed positives for either *Enterobacterales* or *Pseudomonas aeruginosa*. De-identification and data analysis were performed by administrative staff. We included species belonging to the *Enterobacterales* family and *Pseudomonas aeruginosa* due to their prevalence in urinary tract infections, bloodstream infections, and healthcare-associated pneumonia, as well as their increasing resistance to commonly used antimicrobial agents [41–43].

## Statistical analysis

Signals generated from each sensor from enzymatic reaction with TMB substrate were analyzed with three different algorithms for comparison. Before reporting GC ratios, the algorithms first assessed the signal level from the negative and growth controls from each sensor chip. If either control was out of the acceptable range (i.e., greater than 50 nA for the negative control, less than 50 nA for the growth control), the algorithm reported "NC fail" or "GC fail", respectively, indicating substandard quality of a sensor chip or no bacterial growth. If all controls passed the acceptance criteria, the algorithm proceeded to determine the inflection point in the plot of GC ratios against the antibiotic spectrum. The antibiotic concentration corresponding to the inflection point was estimated by two algorithms (Inhibited Growth Cutoff and Maximum Inhibition) and reported as the growth inhibition concentration (GIC). The Inhibited Growth Cutoff method reported the lowest antibiotic concentration with a GC ratio lower than a predetermined cutoff value; the GIC was solely determined by the GC ratio. Initial assessment of the Inhibited Growth Cutoff method used both 0.4 and 0.5 as cutoff values, and the final cutoff value was determined using on-scale strains with an MIC on or one 2-fold dilution above or below the CLSI breakpoints. The Maximum Inhibition method reported the GIC as the lowest antibiotic concentration observed after the maximum GC ratio reduction in the plot. Unlike in the Inhibited Growth Cutoff method, the GIC corresponded to the greatest change in the slope of the response curve as a whole instead of individual GC ratios. For both algorithms, if the GC ratio from the lowest antibiotic concentration tested was less than 0.45, indicating significant growth inhibition, the GIC was reported as less than or equal to this antibiotic concentration. If the GC ratio from the highest antibiotic concentration tested was greater than 0.9, indicating limited growth inhibition, the GIC was reported as greater than this antibiotic concentration. The first level of analysis was qualitative, whereby the antimicrobial efficacy profiles (significant growth, moderate growth, and inhibited growth) derived from the GIC were compared to the corresponding antibiotic susceptibility results (R for resistant, I for intermediate, or S for susceptible) determined by the clinical microbiology lab or CLSI reference methods. The concordance between susceptibilities reported from the GIC and reference susceptibilities was determined by essential and categorical agreement. Essential agreement is an agreement in which a reported MIC value falls within a log2 dilution of the reference MIC from the CDC AR Bank or CLSI broth microdilution. Categorical agreement is an agreement in which a reported S, I, or R interpretation agrees with the reference category

from the CDC AR Bank or CLSI disk diffusion method. Discrepancy rates for the detection of antimicrobial susceptibility were analyzed by very major (vmj), major (maj), and minor errors (min). A vmj, maj, and min are defined as false susceptible reporting (resistant strain reported as susceptible), false resistant reporting (susceptible strain reported as resistant), and misclassification of an intermediate strain (intermediate strain reported as susceptible or resistant), respectively. Any direct-from-specimen antimicrobial efficacy profiles found to be misclassified (i.e., GIC higher than the susceptible breakpoint for a susceptible strain) were retested with both the presented method and microdilution reference method.

The negative control is a direct indicator of the electrochemical reaction between HRP and TMB taking place on the sensor, and the growth control is the quantification of microbial loads in diagnostic specimens without any interference from antimicrobials. Therefore, we assessed the normality of these controls by generating box and whisker plots in S1 and S2 Figs. In S1A and S2 Figs, the distribution of negative controls is positively skewed due to the signal cutoff of our potentiotat reader to detect only amperometric signal from the reduction of oxidized TMB. Although there are few data points that fall in the upper quartile, these data points are still below the negative control maximum of 50 nA.

## Results

There is valid concern about detection sensitivity and matrix interference when developing a direct-from-specimen microbial growth inhibition method that tests directly from clinical specimens rather than an overnight-cultured isolate. Starting directly from unprocessed specimens introduces the challenge of unknown pathogen concentrations ranging from 0 to $> 10^8$ CFU/mL. To address this concern, we established the correlation between the limit of detection (LOD) of the current molecular analysis platform and the assay turnaround time (TAT). Fig 2 illustrates the minimum assay time needed for quantification of RNA transcription at different levels of pathogen concentrations. As shown in Fig 2C, the analyte incubation time for higher target LODs may be significantly reduced, resulting in a TAT of 16 to 36 minutes. Target pathogen enrichment and matrix component removal by centrifugation may be included to achieve mid-level target LODs, resulting in a TAT of 42 to 110 minutes. For low-abundance pathogens and early infection diagnostics, additional viability culture steps may be included to achieve an LOD of $< 10$ CFU/mL with a TAT of 4 to 5.5 hours. The direct-from-specimen antimicrobial efficacy profiling protocol was based on the assay parameters summarized in Fig 2D.

To evaluate the potential impact of urine matrix components on microbial growth inhibition, we tested two sets of contrived samples, one prepared in culture media and the other prepared in negative urine. This initial evaluation was conducted with highly-susceptible *E. coli*

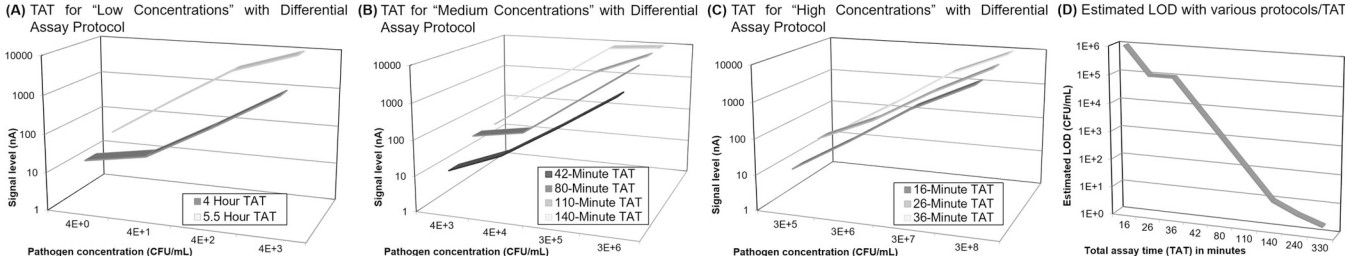

**Fig 2. Calibration curves of configurable ID protocols with various TAT and LODs.** (A) TAT for "low" pathogen concentrations, (B) "medium" pathogen concentrations, (C) "high" pathogen concentrations. (D) Summary of various TAT and LOD. Each plotted point represents 3–4 data points.

(EC69, MIC ≤ 0.06 µg/mL for ciprofloxacin) and highly resistant *K. pneumoniae* (KP79, MIC: >8 µg/mL for ciprofloxacin) isolates from the CDC AR Bank (Fig 3). The goal of the pilot study was to investigate the potential interference in urine; therefore, a higher and more clinically relevant concentration of $1.0 \times 10^7$ CFU/mL was used to contrive the samples. Three antibiotic exposure times of 30, 60, and 90 minutes were tested as primary parameters for optimization. Microbial growth inhibition was analyzed by plotting GC ratios against the ciprofloxacin concentrations tested, which ranged from 0.0625 µg/mL (two 2-fold dilutions below the *Enterobacterales* CLSI susceptible breakpoint) to 4 µg/mL (two 2-fold dilutions above the *Enterobacterales* CLSI resistant breakpoint). As shown in Fig 3A and 3B, all microbial response curves of the resistant *K. pneumoniae* CDC 79 strain were overlapping at the GC ratios near 1.0 (S2 Table), indicating little to no inhibited growth regardless of the exposure time. However, there was a clear trend of inhibited growth, as exhibited by the lower GC ratios, with the susceptible *E. coli* CDC 69 strain; this trend was also more apparent with increasing exposure time or ciprofloxacin concentration. The reported GIC value from the Maximum Inhibition algorithm is listed to the right of each response curve. The bolded GIC value (S strain in MH 30 min, S strain in urine 30 min, S strain in urine 60 min) indicates incorrect categorical susceptibility reporting, which occurred when the exposure time was insufficient. The microbial growth inhibition curves from the contrived urine samples in Fig 3B exhibit characteristics identical to those of the culture media samples in Fig 3A. This similarity suggests that the additional pelleting step performed on the urine samples is sufficient to mitigate the effects of the urine matrix but not harsh enough to put the pathogen into the stationary phase. Additionally, in S1A Fig, the growth controls are clearly separated for each exposure time. Although longer exposure times show a wider distribution, this range may have been caused by the different growth rates of the two included strains, resulting in different signal levels. We also expect there to be a natural dispersion of growth rates within the same strain population. However, these data points are still clearly separated by those from shorter exposure times.

As illustrated in Fig 3A and 3B, it is likely that a shorter antimicrobial exposure time may lead to insignificant growth inhibition of susceptible strains, reducing the separation between susceptible and resistant responses. This phenomenon could potentially lead to more errors in categorical susceptibility reporting without the use of a more sophisticated algorithm. We suspected that a similar reduction in the separation between susceptible and resistant strains would occur if the microbial load were much higher than the standard inoculum density of $5 \times 10^5$ CFU/mL. To evaluate the effects of higher microbial loads and to explore the biological, chemical, and molecular analytical limitations of our assay, we tested contrived urine samples

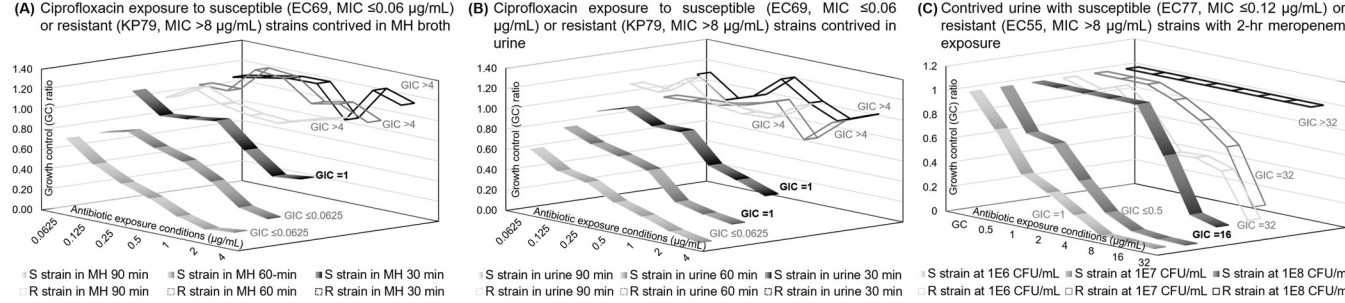

**Fig 3. Investigation of matrix interference components and starting inoculum concentration.** (A) Ciprofloxacin antimicrobial efficacy profiling in MH broth and (B) direct-from-urine ciprofloxacin antimicrobial efficacy profiling using highly susceptible (*E. coli* CDC 69) and highly resistant (*K. pneumoniae* CDC 79) strains from the CDC AR bank. (C) Direct-from-specimen meropenem antimicrobial efficacy profiling with 2-hr exposure for urine with highly susceptible (*E. coli* CDC 77) and resistant strains (*E. coli* CDC 55). Bolded GIC values indicate incorrect categorical susceptibility due to short exposure time (30 or 60 min.) in Fig 3A and 3B or high microbial load ($10^8$ CFU/mL) in Fig 3C. Each plotted point represents 3–4 data points.

prepared at three different microbial loads against a different class of antibiotics (Fig 3C). A shorter antibiotic exposure time of 2 hours was used to assess the separation between resistant and susceptible response curves. Antimicrobial efficacy profiling tests directly from these contrived urine samples were evaluated. Based on the trend of GC ratios along the increasing meropenem concentrations (0.5 to 32 μg/mL), the GIC would be reported as "susceptible" (≤ S-breakpoint of 1 μg/mL for meropenem) for *E. coli* CDC 77 (MIC: ≤ 0.12 μg/mL) and "resistant" (≥ R-breakpoint of 4 μg/mL for meropenem) for *E. coli* CDC 55 (MIC: > 8 μg/mL), which agree with the categorical susceptibilities listed by the CDC AR Bank. The GIC reported from only 2 hours of antimicrobial exposure did not match the MIC value reported from the broth microdilution method, which included a 16-to-24-hour exposure using clinical isolates from an overnight subculture. This disagreement is most likely due to the antimicrobial exposure of the causative pathogen taking place in a different matrix environment (urine vs. agar plate) with different antimicrobial conditions (short vs. long exposure). This study was an initial assessment of the effects of different matrices and testing conditions on categorical susceptibility reporting.

To establish a higher correlation between the MIC and GIC values, it would be necessary to incorporate the impact of the microbial load into the GIC reporting, which is not within the scope of this initial study. With higher contrived concentrations, we expect the inflection point to shift to a higher antimicrobial concentration due to a higher bug-to-drug ratio. Even for susceptible strains, microbial growth can be observed at antibiotic exposure concentrations on or below the susceptible breakpoint if the microbial load is higher than the standard inoculum concentration of 5x10$^5$ CFU/mL.

S1B Fig displays the distribution of growth controls from each inoculum concentration. As the inoculum concentration increases from 10$^6$ to 10$^8$ CFU/mL, more data points become saturated, leading to our hypothesis of a shifting inflection point and inoculum effect on the MIC.

Fig 3C only demonstrated the feasibility to differentiate highly-susceptible from highly-resistant strains, which do not represent all clinical strains; therefore, we wanted to evaluate the growth inhibition curves from on-scale strains containing a MIC on or near the CLSI breakpoints. These strains included *E. coli* CDC 1 with an MIC of 4 μg/mL for gentamicin (on susceptible breakpoint), *E. coli* CDC 85 with an MIC of 1 μg/mL for meropenem (on susceptible breakpoint), and *K. pneumoniae* CDC 80 with an MIC of 0.5 μg/mL for ciprofloxacin (on intermediate breakpoint). To determine if on-scale strains required an exposure time longer than that of highly susceptible and resistant strains, we tested exposure times of 2, 3, and 4 hours. General trends of inhibited growth were observed at 2, 3, and 4 hours for all susceptible-breakpoint strains as shown in Fig 4. The GIC values reported for *E. coli* CDC 1 were at 2 μg/mL for all exposure times; they were one two-fold dilution below the reference MIC (S3 Table). In addition, the categorical susceptibility listed in parentheticals was correctly reported as susceptible. The GIC values for *E. coli* CDC 85 increased from ≤0.5 to 2 μg/mL as the meropenem exposure time increased from 2 to 4 hours. Although the GIC values at all three exposure times were within one 2-fold dilution of the reference MIC for *E. coli* CDC 85, the GIC values with longer exposure times more closely aligned with the MIC value. The goal of this study was to report susceptibility within a much shorter time frame. However, the GIC reporting based on only one response curve with a shorter exposure time of 2 hours was insufficient to differentiate borderline susceptible strains. Therefore, a slightly extended exposure time of 3 hours proved necessary in the case of the CDC 85 strain. We then tested the reproducibility of GIC reporting using two different batches of ciprofloxacin stripwells in Fig 4C and 4D; the GIC reporting was consistent for both batches. The GIC reporting from just two hours of ciprofloxacin exposure was 0.5 μg/mL, which was in agreement with the MIC value from CDC

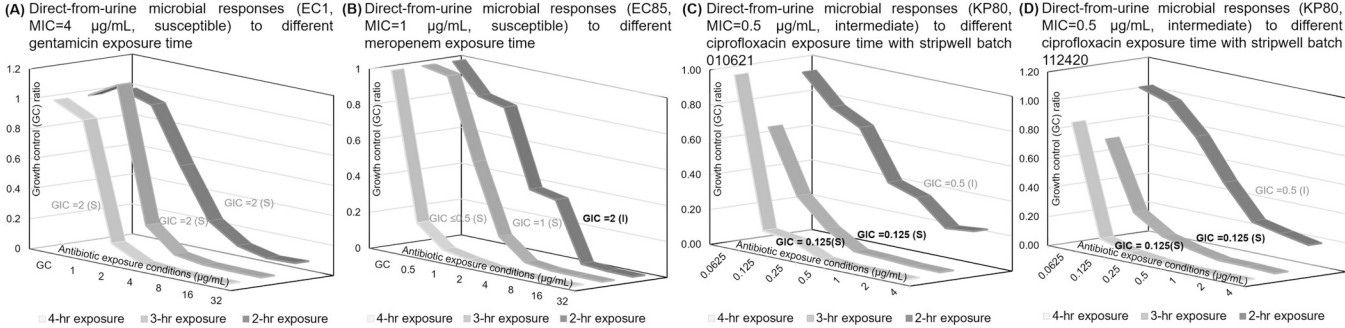

**Fig 4. Varying antibiotic exposure times for direct-from-urine antimicrobial efficacy profiling of on-scale strains for different antibiotics classes.** (A) Gentamicin responses. (B) Meropenem responses. (C, D) Ciprofloxacin responses with different stripwell batches. Bolded GIC values indicate incorrect categorical susceptibility. Each plotted point represents 3–4 data points.

AR Bank database. However, the GIC value transitioned to 0.125 μg/mL with longer exposure times, further signaling the risk of changes in susceptibility reporting when relying on only one response curve. The MIC from the microdilution of *K. pneumoniae* CDC 80 was 0.25 μg/mL, which is within one two-fold dilution from the GIC reporting of all response curves in Fig 4C and 4D.

In S2 Fig, the GC signals are clearly separated for each exposure time. The wider distribution observed for each time may be due to the inclusion of different strains in each dataset, as well as the natural dispersion of growth rates within a single strain population, resulting in different signal levels. Despite this distribution, each exposure time was distinguishable from the others and 3 hours proved to be adequate, unlike 2 and 4 hours, which generated many data points that were either too low or too high (saturated) to observe a clear susceptibility trend.

After demonstrating that 3 hours of antimicrobial exposure may be sufficient for testing on-scale strains, we explored the ability to differentiate bacterial strains with a range of on-scale MIC values on or near the susceptible and resistant breakpoints using a 3-hour exposure time (Fig 5). Fig 5A shows the growth inhibition responses to ciprofloxacin from *E. coli* (EC69:

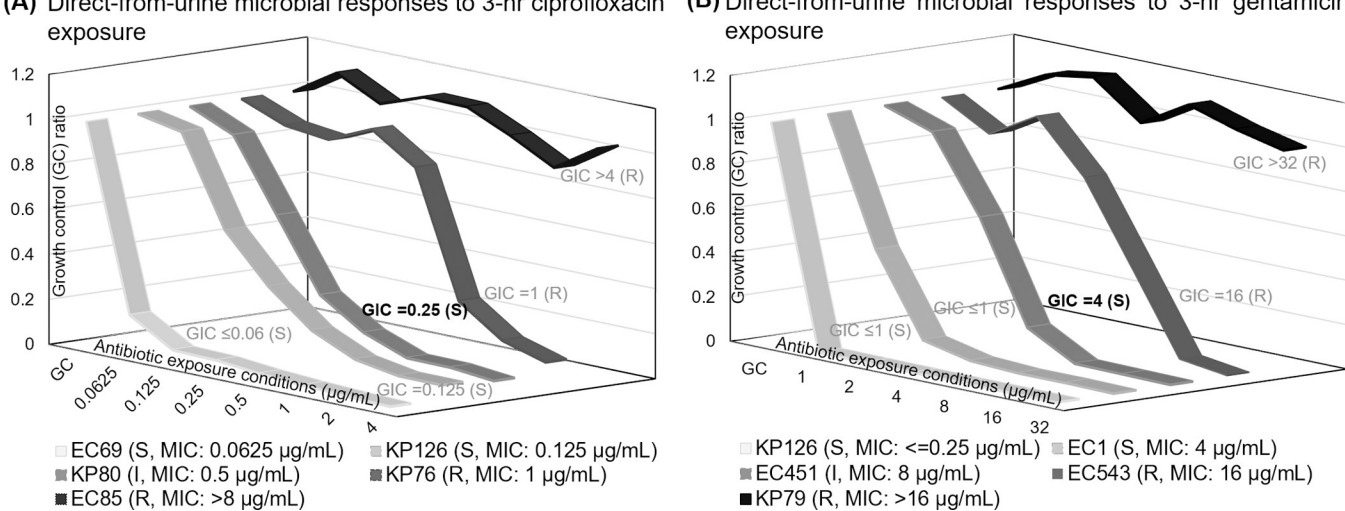

**Fig 5. Direct-from-urine antimicrobial efficacy profiling of pathogens with a range of on-scale MIC values.** All urine specimens contrived at $10^6$ CFU/mL. (A) Ciprofloxacin responses. (B) Gentamicin responses. Bolded GIC values indicate incorrect categorical susceptibility. Each plotted point represents 3–4 data points.

MIC≤0.0625 μg/mL, EC85: MIC> 8μg/mL) and *K. pneumoniae* (KP126: MIC = 0.125μg/mL, KP80: MIC = 0.5 μg/mL, KP76: MIC = 1μg/mL). There is a clear trend of increasing GIC values (≤0.06 μg/mL to >4 μg/mL) that matches the reference MIC values (from ≤0.06 μg/mL to >8 μg/mL), indicating successful distinction of strains with on-scale MICs. Detailed GIC reporting from all three algorithms is displayed in S4 Table. In addition, we evaluated the growth inhibition responses to gentamicin from *K. pneumoniae* (KP126: MIC≤ 0.25μg/mL, KP79: MIC >16 μg/mL) and *E. coli* (EC1: MIC = 4 μg/mL, EC451: MIC = 8 μg/mL, EC543: MIC = 16 μg/mL) in Fig 5B. Similar to Fig 5A, there is a clear trend of increasing GIC values (from ≤1 μg/mL to >32 μg/mL) that matches the reference MIC values (from ≤0.25 μg/mL to >16 μg/mL). Among all susceptible and resistant strains tested in Fig 5, the categorical susceptibility was reported 100% correctly based on the reported GIC value. The two intermediate strains (KP80 for ciprofloxacin and EC451 for gentamicin) were both reported as susceptible, given that both of their GIC values were one two-fold dilution below the reference MIC. We suspected that this incorrect reporting was due to the use of only one response curve. Results leading up to this point in the study suggested the need for a dual-kinetic response curve approach to provide more information on borderline susceptibility such as strains with a MIC on the intermediate breakpoint. Using only one curve resulted in essential agreement between MIC and GIC values, which is acceptable according to CLSI M100 classifications; however, both intermediate strains reported minor errors in categorical susceptibility based on the GIC from one curve [44, 45].

As revealed by the bolded GIC values, categorical susceptibility reporting (susceptible, intermediate, or resistant) may be incorrect if the antimicrobial exposure time is too short (Figs 3A, 3B, and 4B), the microbial load is too high (Fig 3C), or the MIC is on one of the susceptibility breakpoints (Figs 4C, 4D and 5A, 5B). In addition to extending the antimicrobial exposure time—especially for time-dependent antibiotics such as meropenem—we explored the feasibility of a dual-kinetic response approach that would allow us to observe a broader range of microbiological responses. In this approach, we inoculated two stripwells containing the same spectrum of seven antimicrobial concentrations with clinical urine specimens at the original concentration (1x) and at a 10-fold dilution (0.1x). Additionally, to evaluate the correlation between the current GIC reporting algorithm and reference categorical susceptibilities and MIC values throughout the physiological range, we tested a scale of clinically relevant microbial loads for urine ($10^5$ to $10^8$ CFU/mL) in Fig 6. The GIC was calculated from the dual kinetic curves, and the inflection point shifted toward higher antibiotic concentrations in samples with higher microbial loads (S5 Table). In Fig 6B, the growth inhibition curves of 1x and 0.1x of $10^6$ CFU/mL overlapped with each other in the insert graph despite the signal levels of

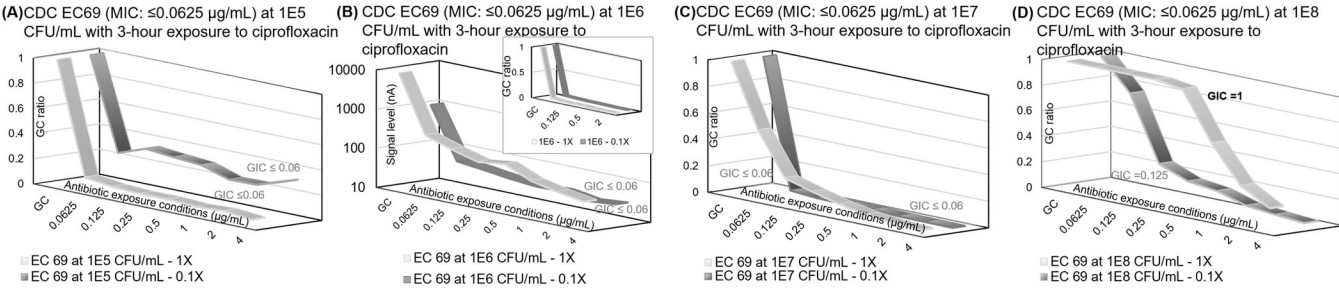

**Fig 6. Direct-from-urine ciprofloxacin antimicrobial efficacy profiling with dual kinetic curves on different contrived urine concentrations.** Dual kinetic curves for *E. coli* CDC 69 with MIC of ≤0.0625 μg/mL at starting sample concentrations of (A) $1.0 \times 10^5$ CFU/mL, (B) $1.0 \times 10^6$ CFU/mL, (C) $1.0 \times 10^7$ CFU/mL, (D) $1.0 \times 10^8$ CFU/mL. The bolded GIC value indicates incorrect categorical susceptibility in individual response curves. Each plotted point represents 3–4 data points.

**Table 1. Ciprofloxacin growth inhibition concentration (GIC) and combined categorical susceptibility reporting for Fig 6.**

| Contrived concentration | 1X response GIC (µg/mL) | 0.1X response GIC (µg/mL) | Combined response GIC (µg/mL) | Combined categorical susceptibility |
|---|---|---|---|---|
| $1.0 \times 10^5$ CFU/mL | $\leq 0.0625$ | $\leq 0.0625$ | $\leq 0.0625$ | Susceptible (categorical agreement) |
| $1.0 \times 10^6$ CFU/mL | $\leq 0.0625$ | $\leq 0.0625$ | $\leq 0.0625$ | Susceptible (categorical agreement) |
| $1.0 \times 10^7$ CFU/mL | $\leq 0.0625$ | $\leq 0.0625$ | $\leq 0.0625$ | Susceptible (categorical agreement) |
| $1.0 \times 10^8$ CFU/mL | 1 | 0.125 | 0.125 | Susceptible (categorical agreement) |

GIC reporting of original sample (1X), dilution (0.1X), and combined dual-curve response of a ciprofloxacin-susceptible strain with an MIC of $\leq 0.0625$ µg/mL.

these two sets of curves being significantly different. It is likely that the similarity in microbial load and symmetry between $10^5$ to $5 \times 10^5$ CFU/mL and from $5 \times 10^5$ to $10^6$ CFU/mL resulted in overlapping GC ratio curves. Fig 6A–6D show the transition of GIC reporting from $\leq 0.0625$ µg/mL (susceptible) to 1 µg/mL (resistant). The categorical susceptibility reporting of "Susceptible" was correct over a range from $10^4$ CFU/mL (0.1x of $10^5$ CFU/mL) to $10^7$ CFU/mL (0.1x of $10^8$ CFU/mL). The GIC value jumped from 0.125 µg/mL (0.1x of $10^8$ CFU/mL) to 1 µg/mL (1x of $10^8$ CFU/mL) as shown in Fig 6D. Similar microbial responses were observed in the rapid ciprofloxacin exposure study with the same *E. coli* CDC 69 strain in Fig 3B; the GIC value jumped from 0.0625 µg/mL (90-min exposure) to 1 µg/mL (30-min and 60-min exposure). The GC signal levels as listed in S5 Table were saturated at 10,000 nA for $10^7$ and $10^8$ CFU/mL; therefore, the reported GIC value is expected to be higher than the MIC values due to the inoculum effect.

The combined categorical susceptibility reporting for the dual-kinetic response approach in Fig 6 is listed in Table 1. We used the Maximum Inhibition algorithm, in which the combined categorical susceptibility is determined by the maximum GC reduction in both microbiological response curves. Specifically, the combined GIC corresponds to the greatest change in the slope of both response curves. Table 1 also includes the individual and combined GIC reporting from all contrived concentrations in Fig 6. Given that the combined categorical susceptibility is determined by the greatest GC ratio reduction in the extended antimicrobial spectrum (all 1x and 0.1x bug-to-drug ratios), it represents the most significant growth inhibition caused by antimicrobial exposure throughout the entire spectrum. Although there was one categorical susceptibility reporting error in the 1x curve in Fig 6D, the reported combined categorical susceptibility was correct for all microbial load conditions. The purpose of the combined GIC reporting in the dual-kinetic-curve approach is to report only the maximum growth inhibition and to discard GIC reporting errors caused by high or low microbial loads.

To evaluate the correlation between the GIC reporting algorithm and reference microbial susceptibility and MIC values throughout the physiological range with other antimicrobial classes, we tested the same set of microbial loads in urine against gentamicin in Fig 7. The GIC

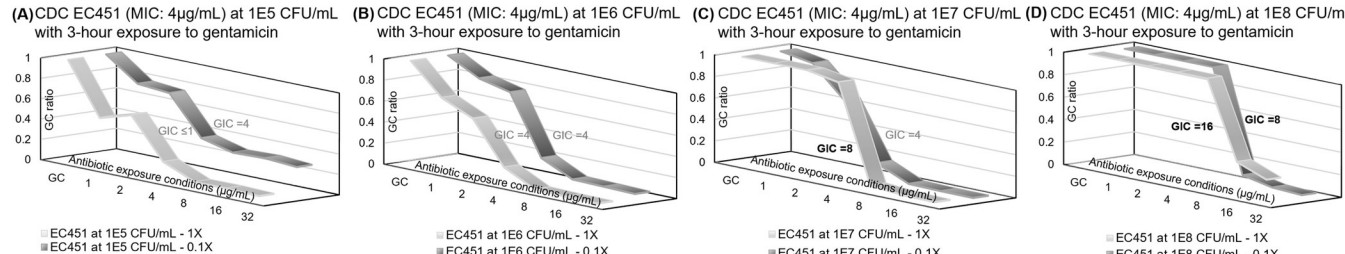

**Fig 7. Direct-from-urine gentamicin antimicrobial efficacy profiling with dual kinetic curves on different contrived urine concentrations.** Dual kinetic curves for *E. coli* CDC 451 with MIC of 4 µg/mL at starting sample concentrations of (A) $1.0 \times 10^5$ CFU/mL, (B) $1.0 \times 10^6$ CFU/mL, (C) $1.0 \times 10^7$ CFU/mL, (D) $1.0 \times 10^8$ CFU/mL. Bolded GIC values indicate incorrect categorical agreement. Each plotted point represents 3–4 data points.

value reported by the Maximum Inhibition algorithm is displayed next to each response curve. GIC reporting from all three algorithms can be found in S6 Table. There is a clear transition in GIC reporting for gentamicin across the range of microbial loads—from ≤1 µg/mL (susceptible) to 16 µg/mL (resistant). The categorical susceptibility reporting of "Susceptible" was correct over a range of $10^4$ CFU/mL (0.1x of $10^5$ CFU/mL) to $10^6$ CFU/mL (0.1x of $10^7$ CFU/mL). A GIC of 8 µg/mL was reported for $10^7$ CFU/mL (1x of $10^7$ CFU/mL and 0.1x of $10^8$ CFU/mL), which is one dilution above the reference MIC of 4 µg/mL. This disagreement between GIC and MIC is acceptable for essential agreement but is a minor error for categorical agreement. The GIC reporting of 16 µg/mL from $10^8$ CFU/mL was a major error. Similar to the results of ciprofloxacin, the GC signal levels, as listed in S6 Table, were saturated at 10,000 nA for $10^7$ and $10^8$ CFU/mL; therefore, the reported GIC value is expected to be higher than the reference MIC value due to the inoculum effect.

Table 2 is a summary of the individual and combined GIC reporting from all contrived concentrations in Fig 7. The MIC of *E. coli* CDC 451 is listed by the CDC as 8 µg/mL, indicating intermediate susceptibility, but our microdilution indicated a MIC of 4 µg/mL, which would be categorically classified as susceptible. Therefore, we used the MIC of 4 µg/mL for reference, as it was obtained with the reference microdilution method. Without adjusting for the inoculum effect, the maximum growth inhibition would indicate a combined GIC of 8 µg/mL for both Fig 7C and 7D. However, due to the signal level saturation observed at the growth control and low antibiotic concentrations (1 and 2 µg/mL in 1x curve in Fig 7C, 1–8 µg/mL in 1x curve in Fig 7D, 1–4 µg/mL in 0.1x curve in Fig 7D), we adjusted the combined GIC to account for the inoculum effect. The electrochemical current reading is set to saturate at 10,000 nA to maximize the resolution at lower current readings around the limit of detection, so the reading would be saturated if the starting microbial load were too high. The reported GIC was adjusted one dilution down for every antibiotic concentration reported at a saturated signal level. As a result, the combined GIC reporting from Fig 7C and 7D was adjusted from 8 µg/mL to 4 µg/mL. In comparison to the microdilution MIC, there were three categorical susceptibility reporting errors in the single response curves in Fig 7C and 7D. After adjusting for the saturated signal level, the combined categorical susceptibility of both response curves was correct for all microbial load conditions.

Similar results were observed for the same study using meropenem in Fig 8. The reported GIC transitioned from ≤0.5 µg/mL (susceptible) to 32 µg/mL (resistant). The categorical susceptibility reporting of "Resistant" was correct over a range of $10^5$ CFU/mL to $10^8$ CFU/mL. A GIC of ≤0.5µg/mL was reported for $10^4$ CFU/mL (0.1x of $10^5$ CFU/mL in Fig 8A) and resulted in a very major error for categorical agreement. However, the GC signal level listed in S7 Table was 39 nA, which indicated insufficient microbial growth and was reported as "GC fail". No GIC value was reported in the case of GC failures (<50 nA).

Table 3 is a summary of the individual and combined GIC reporting for Fig 8. Similar to the CDC 451 strain, the MIC value of *K. pneumoniae* CDC 79 is listed by the CDC as 8 µg/mL,

**Table 2. Gentamicin growth inhibition concentration (GIC) and combined categorical susceptibility reporting for Fig 7.**

| Contrived concentration | 1X response GIC (µg/mL) | 0.1X response GIC (µg/mL) | Combined response GIC (µg/mL) | Combined categorical susceptibility |
|---|---|---|---|---|
| 1.0x$10^5$ CFU/mL | ≤1 | 4 | 4 | Susceptible (categorical agreement) |
| 1.0x$10^6$ CFU/mL | 4 | 4 | 4 | Susceptible (categorical agreement) |
| 1.0x$10^7$ CFU/mL | 8 → 4 | 4 | 4 | Susceptible (categorical agreement) |
| 1.0x$10^8$ CFU/mL | 16 → 8 | 8 → 4 | 4 | Susceptible (categorical agreement) |

GIC reporting of original sample (1X), dilution (0.1X), and combined dual-curve response of a gentamicin susceptible strain with an MIC of 4 µg/mL.

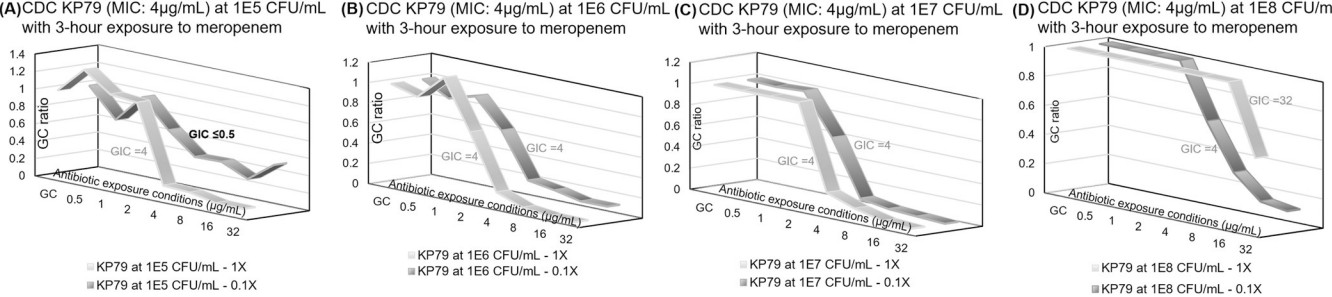

**Fig 8. Direct-from-urine meropenem antimicrobial efficacy profiling dual kinetic curves for different starting sample concentrations.** Dual kinetic curves for *K. pneumoniae* CDC 79 with MIC of 4 μg/mL at starting sample concentrations of (A) $10^5$ CFU/mL, (B) $10^6$ CFU/mL, (C) $10^7$ CFU/mL, (D) $10^8$ CFU/mL. Bolded GIC values indicate incorrect categorical susceptibility. Each plotted point represents 3–4 data points.

but our microdilution indicated the MIC was 4 μg/mL. We used the microdilution result as the reference. Without adjusting for the inoculum effect, the maximum growth inhibition would result in a combined GIC of 0.5 and 32 μg/mL for Fig 8A and 8D, respectively. However, the combined GIC was adjusted due to growth control failure in Fig 8A and saturated signal level at the growth control and five antibiotic concentrations in Fig 8D. There was initially only one categorical susceptibility reporting error in Fig 8A, but it was not reported due to GC failure. The combined categorical susceptibility was correct for all microbial load conditions.

After initial validation of the presented antimicrobial efficacy profiling method using CDC clinical strains, we conducted a pilot feasibility study on blinded urine specimens from NYPQ. De-identified remnant clinical specimens were shipped overnight to GeneFluidics for testing as described above, and the summary of combined categorical susceptibility is detailed in Table 4. Sample #7 was positive for *P. aeruginosa* but when tested with the assay produced a GC failure. Subculture of NYPQ sample #7 on a Chromagar plate exhibited two separate strains, indicating that the original specimen may have contained a polymicrobial infection or was contaminated during sample collection or testing. For specimens containing multiple organisms, species-specific susceptibility reporting would require the pathogen identification sensor chip with complementary oligonucleotide probes for each target pathogen, which is outside the scope of this study. The categorical susceptibilities of the remaining nine specimens were reported correctly, resulting in 100% categorical agreement with the susceptibilities reported by NYPQ. All individual and combined GIC reports are listed in S8 Table. NYPQ's AST panel tests levofloxacin (LEV) instead of ciprofloxacin (CIP) for the class of fluoroquinolones; therefore, the GIC reporting of CIP susceptibility for Samples 1, 4, and 6 was compared to the categorical susceptibility interpreted from the reference broth microdilution result. Levofloxacin is generally less effective than ciprofloxacin against Gram-negative pathogens, as explained in the literature [46, 47]. If a pathogen is susceptible to levofloxacin, it may not be

**Table 3. Meropenem growth inhibition concentration (GIC) and combined categorical susceptibility reporting for Fig 8.**

| Contrived concentration | 1X response GIC (μg/mL) | 0.1X response GIC (μg/mL) | Combined response GIC (μg/mL) | Combined categorical susceptibility |
|---|---|---|---|---|
| $1.0 \times 10^5$ CFU/mL | 4 | ≤0.5 | 4 | Resistant (categorical agreement) |
| $1.0 \times 10^6$ CFU/mL | 4 | 4 | 4 | Resistant (categorical agreement) |
| $1.0 \times 10^7$ CFU/mL | 4 | 4 | 4 | Resistant (categorical agreement) |
| $1.0 \times 10^8$ CFU/mL | 32 → 20 | 4 | 20 | Resistant (categorical agreement) |

GIC reporting of original sample (1X), dilution (0.1X), and combined dual-curve response of a meropenem-resistant strain with an MIC of 4 μg/mL.

**Table 4. Summary of direct-from-urine antimicrobial efficacy profiling using de-identified remnant urine specimens from NYPQ.**

| Sample Code | Organism | 1X response GIC (μg/mL) | 0.1X response GIC (μg/mL) | Combined response GIC (μg/mL) | NYPQ reported susceptibility or microdilution (MIC: μg/mL) | Combined categorical susceptibility |
|---|---|---|---|---|---|---|
| 0001 | Citrobacter koseri | 0.125 | <0.06 | <0.06 | CIP susceptible (≤0.0625) | Susceptible (categorical agreement) |
| 0002 | Escherichia coli | <0.5 | 0.5 | <0.5 | MEM susceptible (≤0.25) | Susceptible (categorical agreement) |
| 0003 | Enterobacter cloacae complex | 16 | 16 | 16 | GEN resistant (> = 16) | Resistant (categorical agreement) |
| 0004 | Escherichia coli | 0.5 | 0.25 | 0.5 | CIP susceptible (0.5) | Intermediate (categorical agreement) |
| 0005 | Serratia marcescens | <0.5 | No growth | <0.5 | MEM susceptible (≤0.25) | Susceptible (categorical agreement) |
| 0006 | Klebsiella pneumoniae | >4 | >4 | >4 | CIP resistant (>4) | Resistant (categorical agreement) |
| 0007 | Pseudomonas aeruginosa | No growth | No growth | | MEM resistant (> = 32) | No growth |
| 0008 | Proteus penneri | <1 | <1 | <1 | GEN susceptible (<1) | Susceptible (categorical agreement) |
| 0009 | Citrobacter koseri | <1 | <1 | <1 | GEN susceptible (≤1) | Susceptible (categorical agreement) |
| 0010 | Escherichia coli | 32 | 32 | 32 | GEN resistant (> = 16) | Resistant (categorical agreement) |

Each antimicrobial condition tested represents 3–4 data points.

susceptible to ciprofloxacin, as demonstrated in Sample 4. However, if a pathogen is resistant to ciprofloxacin, it is likely to be resistant to levofloxacin, as observed in Sample 6.

## Discussion

Although recent technologies have allowed PCR-based pathogen identification to be performed in fewer than 30 minutes, there is currently no phenotypic AST that can be performed within a reasonable time frame—specifically, in hours—directly from clinical samples in clinical microbiology laboratory settings. Schoepp et al. demonstrated a benchtop digital LAMP quantification method that measured the phenotypic response of *E. coli* in clinical urine samples and presented AST results after a 15-minute antibiotic exposure. However, only highly-resistant or susceptible strains with rapid doubling times were selected for testing [48]. For pathogens with an on-scale MIC or a longer doubling time, an extended antibiotic-exposure incubation is necessary. Khazaei et al. demonstrated that quantifying changes in RNA signatures instead of DNA replication resulted in significant shifts (>4-fold change) in transcription levels within 5 minutes of antibiotic exposure [37, 49]. However, there was a wide range of control:treated (C:T) ratio dispersion from highly susceptible strains with MICs at least seven 2-fold dilutions below the resistant breakpoint. With 8 strains of the same MIC (0.015 μg/mL) and one strain with an MIC only 2-fold above (0.03 μg/mL), the C:T ratio can change from 2 to 6, but the C:T ratio separation between resistant and susceptible populations is only roughly 0.4. The aforementioned study demonstrates the limitation observed in clinical settings where not all susceptible strains have an extremely low MIC.

Doern noted that although the concept of using unprocessed clinical specimens as inoculum for direct-from-specimen AST or antimicrobial efficacy profiling is appealing, there are significant challenges to this approach [1]. The first challenge he mentioned was accommodating clinical specimens with unknown organism concentrations that may be significantly

higher or lower than the standardized inoculum concentration used in most growth-based susceptibility tests. In a proof-of-concept study by Mezger et al., urine was used as an inoculum for rapid AST, in which a 120-minute antimicrobial exposure was performed, followed by quantitative PCR [50]. Although pilot experiments demonstrated *E. coli* susceptibility to ciprofloxacin and trimethoprim within 3.5 hours, the susceptibility profiling algorithm was not correlated to CLSI M100 categorical reporting. In our method, we attempted to address this second challenge of providing susceptibility profiling equivalent to AST performed in a clinical microbiology lab (>95% categorical agreement) by assessing susceptibility response dynamic trends at three different bug/drug ratios. This was done by inoculating the raw specimens in two dilutions as detailed above. The third challenge is the need to ensure pathogens are isolated from clinical samples to allow for retesting, confirmation of phenotypic testing (e.g., AST), polymicrobial testing, or epidemiological studies. This challenge will be addressed by setting aside the remainder of specimens for QC or archiving purposes.

Despite being recognized as the standard quantitative index of antimicrobial potency, the MIC is subject to several limitations, the first of which is a long antimicrobial exposure time of 16 to 20 hours. Furthermore, it requires a standard inoculum concentration of $5x10^5$ CFU/mL, rendering it insufficient to test a low initial bacterial inoculum (i.e., 3 to 5 colonies usually in the absence of resistant populations). Lastly, it utilizes constant, or static, antibiotic concentrations [51]. Therefore, the MIC provides no information on the time-course of bacterial killing or of emergence of resistance [52–56]. Several static and dynamic *in vitro* and *in vivo* infection model studies have been performed to analyze and interpret *in vitro* efficacy results of antimicrobial drugs as an alternative to MIC reporting [56–61]. These experimental models provide a wealth of time-course data on bacterial growth and killing but have not been adopted into a diagnostic test directly from clinical specimens [62].

An ideal growth inhibition spectrum can fit concentration-responses in a sigmoidal curve that is symmetrical about its inflection point and flattened on both ends with statistical fluctuations, as shown in Figs 6–8. The left plateau represents insignificant growth inhibition under antibiotic exposure below the MIC, and the right plateau represents significant growth inhibition above the MIC. The inflection point indicates the concentration at which antimicrobial potency lies midway between non-inhibited growth (left plateau) and complete inhibited growth (right plateau); the slope of the tangent to the curve at the inflection point is a measure of the antimicrobial intensity.

With increasing concentrations of antibiotic in each well, the effectiveness of the antibiotic increases and lowers the rate of pathogen viability. This behavior is reflected in the growth control ratio, which would be negatively correlated with the instantaneous mortality rate. Therefore, the antimicrobial concentration at the inflection point, or GIC, will likely increase when the microbial load in the clinical specimen is higher. This concept is exemplified in the literature [63–66]. Based on this hypothesis, we developed a direct-from-specimen microbial growth inhibition test that utilizes two dilutions of unprocessed clinical specimens (1x and 0.1x) as inoculums for two antibiotic exposure stripwells, each containing one GC well and the same range of seven antibiotic concentrations. The resulting response curves are used to visualize the microbial growth inhibition spectrum. As the drug concentration increases, the probability of drug molecules reaching a lethal concentration increases as a function modeled by a smooth sigmoidal curve. Considering the unknown microbial load in clinical specimens, the coverage of this spectrum is designed to capture the inflection point within the entire range of physiological conditions. The GC well of each stripwell serves two purposes: assist in GIC adjustment based on the microbial load under no antibiotics and provide quality control to eliminate the data set if there is no growth due to a microbial load below the limit of detection. In this study, we developed a tentative algorithm that aims to identify the antibiotic

concentration at the inflection point and adjust this inflection point based on microbial load determined by the GC signal level; the reported GICs were compared to the MIC obtained from reference methods or FDA-cleared systems.

## Supporting information

**S1 Fig. Signal distribution of negative and growth controls for Fig 3.** (A) Negative controls for Fig 3 and growth controls for Fig 3A and 3B, (B) Growth controls for Fig 3C.
(TIF)

**S2 Fig. Signal distribution of negative and growth controls for Fig 4.**
(TIF)

**S3 Fig. Signal distribution of negative and growth controls for Figs 3 and 4 combined.**
(TIF)

**S1 Table. Clinical isolate counts with strain # and antimicrobial tested (MIC).** On-scale strains in bold.
(PDF)

**S2 Table. A. GIC reporting values for Fig 3A.** Ciprofloxacin GIC reporting with three algorithms for *E. coli* CDC 69 with a MIC of $\leq$ 0.0625 μg/mL and *K. pneumoniae* CDC 79 with an MIC of >8 μg/mL for Fig 3A. **B. GIC reporting values for Fig 3B.** Ciprofloxacin GIC reporting with three algorithms for *E. coli* CDC 69 with a MIC of $\leq$ 0.0625 μg/mL and *K. pneumoniae* CDC 79 with a MIC of >8 μg/mL for Fig 3B. **C. GIC reporting values for Fig 3C.** Meropenem GIC reporting with three algorithms for *E. coli* CDC 77 with a MIC of $\leq$ 0.12 μg/mL and *E. coli* CDC 55 with an MIC of > 8 μg/mL.
(PDF)

**S3 Table. A. GIC reporting values for Fig 4A.** Gentamicin GIC reporting with three algorithms for *E. coli* CDC 1 with an MIC of 4 μg/mL. **B. GIC reporting values for Fig 4B.** Meropenem GIC reporting with three algorithms for *E. coli* CDC 85 with an MIC of 1 μg/mL. **C. GIC reporting values for Fig 4C.** Ciprofloxacin GIC reporting with three algorithms for *K. pneumoniae* CDC 80 with an MIC of 0.5 μg/mL. **D. GIC reporting values for Fig 4D.** Ciprofloxacin GIC reporting with three algorithms for *K. pneumoniae* CDC 80 with an MIC of 0.5 μg/mL.
(PDF)

**S4 Table. A.** GIC reporting values in three algorithms for Fig 5A. **B.** GIC reporting values in three algorithms for Fig 5B.
(PDF)

**S5 Table. GIC reporting values for Fig 6.** Ciprofloxacin GIC reporting with three algorithms for *E. coli* CDC 69 with a MIC of $\leq$ 0.0625 μg/mL.
(PDF)

**S6 Table. GIC reporting values for Fig 7.** Gentamicin GIC reporting with three algorithms for *E. coli* CDC 451 with a MIC of 4 μg/mL.
(PDF)

**S7 Table. GIC reporting values for Fig 8.** Meropenem GIC reporting with three algorithms for *K. pneumoniae* CDC 79 with a MIC of 4 μg/mL.
(PDF)

**S8 Table. GIC reporting of NYPQ blinded clinical specimens.**
(PDF)

**S1 Protocol. Molecular quantification of 16S rRNA with multiplexed-electrochemical biosensors through enzymatic amplification of redox current without using PCR.**
(DOCX)

## Author Contributions

**Conceptualization:** Jade Chen, Amelia Kung, Michael Tomasek, Dakai Liu.

**Data curation:** Jade Chen, Su Su Soe San, Amelia Kung, Michael Tomasek.

**Formal analysis:** Jade Chen, Amelia Kung, Michael Tomasek.

**Investigation:** Jade Chen, Michael Tomasek, Vincent Gau.

**Methodology:** Jade Chen, Michael Tomasek, Vincent Gau.

**Project administration:** Vincent Gau.

**Resources:** Dakai Liu, William Rodgers.

**Supervision:** Dakai Liu, William Rodgers, Vincent Gau.

**Validation:** Jade Chen, Michael Tomasek.

**Visualization:** Michael Tomasek, Dakai Liu.

**Writing – original draft:** Jade Chen, Michael Tomasek.

**Writing – review & editing:** Michael Tomasek, Vincent Gau.

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
