## [Decision Letter · Decision Letter 0]

8 Jul 2021

PONE-D-21-08657

Direct-from-specimen microbial growth inhibition spectrums under antibiotic exposure and comparison to conventional antimicrobial susceptibility testing

PLOS ONE

Dear Dr. Gau,

Thank you for submitting your manuscript to PLOS ONE. After careful consideration, we feel that it has merit but does not fully meet PLOS ONE’s publication criteria as it currently stands. Therefore, we invite you to submit a revised version of the manuscript that addresses the points raised during the review process.

We look forward to receiving your revised manuscript.

Kind regards,

Arumugam Sundaramanickam, PhD

Academic Editor

PLOS ONE

Journal Requirements:

- https://journals.asm.org/doi/10.1128/jcm.01999-17?permanently=true

The text that needs to be addressed involves the second paragraph of the Discussions section.

In your revision ensure you cite all your sources (including your own works), and quote or rephrase any duplicated text outside the methods section. Further consideration is dependent on these concerns being addressed.

Reviewers' comments:

Reviewer's Responses to Questions

**Comments to the Author**

1. Is the manuscript technically sound, and do the data support the conclusions?

Reviewer #1: Partly

Reviewer #2: Partly

2. Has the statistical analysis been performed appropriately and rigorously? 

Reviewer #1: No

Reviewer #2: N/A

3. Have the authors made all data underlying the findings in their manuscript fully available?

Reviewer #1: Yes

Reviewer #2: Yes

4. Is the manuscript presented in an intelligible fashion and written in standard English?

Reviewer #1: No

Reviewer #2: Yes

5. Review Comments to the Author

Reviewer #1: Summary of the manuscript:

I would like to appreciate the authors for their efforts, overall ideology and execution of the work is commendable. But language of the manuscript is not upto the mark, it has lots of grammatical errors and improper sentence formations. The manuscript needs an extensive review of the English language, hence this manuscript can’t be considered for publication in the same format.

Specific corrections

Avoid general statements, results and discussions in the methodology part, it can be included in respective sections of the manuscript.

Authors may avoid some unsound words in manuscript such as, very, see etc.

Authors may provide figures with high clarity

Reviewer #2: The manuscript “Direct-from-specimen microbial growth inhibition spectrums under antibiotic exposure and comparison to conventional antimicrobial susceptibility testing” proposes an antimicrobial efficacy profiling method that can be performed in hours directly from clinical urine specimens. The study seems very interesting, and clearly advances the current science. Although authors have worked on an interesting area of research, the figure quality is too poor to judge the manuscript. I am unable to read anything; the printing was also not helpful. Please submit higher resolution images for proper review. Please avoid abbreviations in the abstract. Also, adding a graphical abstract will be helpful for the readers of Plos one.

6. PLOS authors have the option to publish the peer review history of their article (what does this mean?). If published, this will include your full peer review and any attached files.

Reviewer #1: No

Reviewer #2: No

---

## [Author Response · Author response to Decision Letter 0]

29 Jul 2021

Response to Reviewers

PONE-D-21-08657

Dear Editor,

We would like to thank you and the reviewers for taking the time to consider our work and provide feedback. We appreciate the reviewers’ detailed feedback, which has proved to be helpful in improving the quality of our manuscript. We have adjusted our manuscript accordingly and addressed each comment in italics with references to the clean copy of the revised manuscript. 

Journal Requirements:

Response: We have ensured our manuscript meets journal requirements: figure and table references, figure and table captions and legends, section headings, etc.

- https://journals.asm.org/doi/10.1128/jcm.01999-17?permanently=true

The text that needs to be addressed involves the second paragraph of the Discussions section.

In your revision ensure you cite all your sources (including your own works), and quote or rephrase any duplicated text outside the methods section. Further consideration is dependent on these concerns being addressed.

Response: We have amended lines 482-490 to address the overlapping text. We have also checked all citations for other instances of overlapping text.

Comments to the Author

1. Is the manuscript technically sound, and do the data support the conclusions?

Reviewer #1: Partly

Reviewer #2: Partly

Response: We have added the number of data points for each graph in each figure caption. We have also revised the statistical analysis section (line 170) to provide more details.

2. Has the statistical analysis been performed appropriately and rigorously? 

Reviewer #1: No

Reviewer #2: N/A

Response: We have added the number of data points for each graph in each figure caption. We have also revised the statistical analysis section (line 170) to provide more details.

3. Have the authors made all data underlying the findings in their manuscript fully available?

Reviewer #1: Yes

Reviewer #2: Yes

4. Is the manuscript presented in an intelligible fashion and written in standard English?

Reviewer #1: No

Reviewer #2: Yes

Response: We have made changes throughout the manuscript to more clearly present our research findings and enhance the flow of content.

5. Review Comments to the Author

Reviewer #1: Summary of the manuscript:

I would like to appreciate the authors for their efforts, overall ideology and execution of the work is commendable. But language of the manuscript is not upto the mark, it has lots of grammatical errors and improper sentence formations. The manuscript needs an extensive review of the English language, hence this manuscript can’t be considered for publication in the same format.

Response: We have made changes throughout the manuscript to more clearly present our research findings and enhance the flow of content.

Specific corrections

Avoid general statements, results and discussions in the methodology part, it can be included in respective sections of the manuscript.

Response: We have removed such statements from the methods section.

Authors may avoid some unsound words in manuscript such as, very, see etc.

Response: We have made changes throughout the manuscript to address this concern.

Authors may provide figures with high clarity

Response: We have improved the quality of the figures.

Reviewer #2: The manuscript “Direct-from-specimen microbial growth inhibition spectrums under antibiotic exposure and comparison to conventional antimicrobial susceptibility testing” proposes an antimicrobial efficacy profiling method that can be performed in hours directly from clinical urine specimens. The study seems very interesting, and clearly advances the current science. Although authors have worked on an interesting area of research, the figure quality is too poor to judge the manuscript. I am unable to read anything; the printing was also not helpful. Please submit higher resolution images for proper review. Please avoid abbreviations in the abstract. Also, adding a graphical abstract will be helpful for the readers of Plos one.

Response: We have removed all abbreviations in the abstract. We have improved the quality of the figures and the surrounding text. We have included a graphical abstract (Fig 1) illustrating the presented method in line 76.

---

## [Decision Letter · Decision Letter 1]

6 Dec 2021

PONE-D-21-08657R1

Direct-from-specimen microbial growth inhibition spectrums under antibiotic exposure and comparison to conventional antimicrobial susceptibility testing

PLOS ONE

Dear Dr. Gau,

Thank you for submitting your manuscript to PLOS ONE. The 2nd reviewer has raised a few minor issues regarding the literature discussed and statistical analyses performed in the manuscript. Therefore, we invite you to submit a revised version of the manuscript that addresses these minor points.

Please submit your revised manuscript within a month. If you will need more time than this to complete your revisions, please reply to this message or contact the journal office at plosone@plos.org. Please include the following items when submitting your revised manuscript:

We look forward to receiving your revised manuscript.

Kind regards,

Mehmet A Orman

Academic Editor

PLOS ONE

Journal Requirements:

Reviewers' comments:

Reviewer's Responses to Questions

**Comments to the Author**

1. If the authors have adequately addressed your comments raised in a previous round of review and you feel that this manuscript is now acceptable for publication, you may indicate that here to bypass the “Comments to the Author” section, enter your conflict of interest statement in the “Confidential to Editor” section, and submit your "Accept" recommendation.

Reviewer #1: All comments have been addressed

Reviewer #2: (No Response)

2. Is the manuscript technically sound, and do the data support the conclusions?

Reviewer #1: Yes

Reviewer #2: Partly

3. Has the statistical analysis been performed appropriately and rigorously? 

Reviewer #1: Yes

Reviewer #2: No

4. Have the authors made all data underlying the findings in their manuscript fully available?

Reviewer #1: Yes

Reviewer #2: Yes

5. Is the manuscript presented in an intelligible fashion and written in standard English?

Reviewer #1: Yes

Reviewer #2: Yes

6. Review Comments to the Author

Reviewer #1: All the comments have been addressed by authors, I would like to recommend this manuscript for the publication.

Reviewer #2: The paragraph “Even though antibiotics do not directly affect the SARS-CoV-2 respiratory virus responsible for the COVID-19 48 pandemics, physicians are administering many more antibiotics than normal when treating COVID-19 patients 49 [2]. As published in the New England Journal of Medicine, 58.0% of the surveyed 1,099 COVID-19 patients 50 received intravenous antibiotic therapy in China, whereas 35.8% received oseltamivir antiviral therapy [3]. The 51 appearing surge in antibiotic use is reflected in the higher percentages of COVID-19 patients with severe 52 conditions and pediatric patients (85% in a multicenter pediatric COVID-19 study [4]) receiving antibiotic 53 therapies. The World Health Organization warned that the use of antibiotic therapy may lead to higher bacterial 54 resistance rates and increase the burden of the pandemic [5]. As viral respiratory infections often lead to bacterial 55 pneumonia, physicians may struggle to identify the causative pathogen in lung infections. A recent study by Zhou 56 et al. [6] found that 15% of 191 hospitalized COVID-19 patients, as well as 50% of the 54 non-survivors, acquired 57 bacterial infections. Major outbreaks of other respiratory viruses illustrated the same concern. For example, the 58 majority of deaths from the 1918 flu showed autopsy results consistent with bacterial pneumonia, and up to half 59 of the 300,000 non-survivors of the 2009 H1N1 flu were confirmed to have died from pneumonia [7-8]. Therefore, 60 a shorter time to rule out certain antibiotic options by detecting microbial growth under such conditions may 61 provide physicians with valuable information before the availability of conventional AST results,” contains some irrelevant information and it must be shortened.

Where statistical analyses are described, please specify what formal tests for normality were used to assess data distribution. All data should be subject to tests for normality. Data that do not exhibit a normal/Gaussian distribution should be analyzed via a non-parametric equivalent.

7. PLOS authors have the option to publish the peer review history of their article (what does this mean?). If published, this will include your full peer review and any attached files.

Reviewer #1: No

Reviewer #2: No

---

## [Author Response · Author response to Decision Letter 1]

22 Dec 2021

Dear Editor, 

We would like to thank you and the reviewers for considering our manuscript for publication. We appreciate the time and effort taken to provide feedback. Although comments were minor, we believe they were helpful to ensure the best version of our manuscript. We have addressed all comments below in italics and have referenced the manuscript with tracked changes in our responses. 

Reviewers' comments: 

Reviewer's Responses to Questions 

Comments to the Author 

1. If the authors have adequately addressed your comments raised in a previous round of review and you feel that this manuscript is now acceptable for publication, you may indicate that here to bypass the “Comments to the Author” section, enter your conflict of interest statement in the “Confidential to Editor” section, and submit your "Accept" recommendation. 

Reviewer #1: All comments have been addressed 

Reviewer #2: (No Response) 

2. Is the manuscript technically sound, and do the data support the conclusions? 

Reviewer #1: Yes 

Reviewer #2: Partly 

Response: We have added statistical analysis in the supporting information. Please see response below. 

3. Has the statistical analysis been performed appropriately and rigorously? 

Reviewer #1: Yes 

Reviewer #2: No 

Response: We have added statistical analysis in the supporting information. Please see response below. 

4. Have the authors made all data underlying the findings in their manuscript fully available? 

Reviewer #1: Yes 

Reviewer #2: Yes 

5. Is the manuscript presented in an intelligible fashion and written in standard English? 

Reviewer #1: Yes 

Reviewer #2: Yes 

6. Review Comments to the Author 

Reviewer #1: All the comments have been addressed by authors, I would like to recommend this manuscript for the publication. 

Response: We thank you for your time and support. 

Reviewer #2: The paragraph “Even though antibiotics do not directly affect the SARS-CoV-2 respiratory virus responsible for the COVID-19 48 pandemics, physicians are administering many more antibiotics than normal when treating COVID-19 patients 49 [2]. As published in the New England Journal of Medicine, 58.0% of the surveyed 1,099 COVID-19 patients 50 received intravenous antibiotic therapy in China, whereas 35.8% received oseltamivir antiviral therapy [3]. The 51 appearing surge in antibiotic use is reflected in the higher percentages of COVID-19 patients with severe 52 conditions and pediatric patients (85% in a multicenter pediatric COVID-19 study [4]) receiving antibiotic 53 therapies. The World Health Organization warned that the use of antibiotic therapy may lead to higher bacterial 54 resistance rates and increase the burden of the pandemic [5]. As viral respiratory infections often lead to bacterial 55 pneumonia, physicians may struggle to identify the causative pathogen in lung infections. A recent study by Zhou 56 et al. [6] found that 15% of 191 hospitalized COVID-19 patients, as well as 50% of the 54 non-survivors, acquired 57 bacterial infections. Major outbreaks of other respiratory viruses illustrated the same concern. For example, the 58 majority of deaths from the 1918 flu showed autopsy results consistent with bacterial pneumonia, and up to half 59 of the 300,000 non-survivors of the 2009 H1N1 flu were confirmed to have died from pneumonia [7-8]. Therefore, 60 a shorter time to rule out certain antibiotic options by detecting microbial growth under such conditions may 61 provide physicians with valuable information before the availability of conventional AST results,” contains some irrelevant information and it must be shortened. 

Response: We have removed irrelevant information and shortened the paragraph in lines 47-61. 

Where statistical analyses are described, please specify what formal tests for normality were used to assess data distribution. All data should be subject to tests for normality. Data that do not exhibit a normal/Gaussian distribution should be analyzed via a non-parametric equivalent. 

Response: The main premise of this paper is to demonstrate the feasibility of a direct-from-specimen AST that does not require clinical isolates as most current FDA-approved and CLSI methods do. A larger sample size for testing of clinical samples would be our next step to further validate the clinical utility of the presented method, and the normality test would then be essential for that future study. However, we have taken your suggestion and added a paragraph to the statistical analysis section (lines 205-211) to explain our normality tests for the negative and growth controls. Additionally, we reference and briefly explain each supplemental figure in lines 251-255, 282-284, and 317-322. 

In the future study using a larger sample size of blinded clinical samples, the microbial load distribution would not be predictable and therefore not normal, as samples could be negative, high positive, or weak positive (>80% of culture samples are reported negative). 

7. PLOS authors have the option to publish the peer review history of their article (what does this mean?). If published, this will include your full peer review and any attached files. 

Do you want your identity to be public for this peer review? For information about this choice, including consent withdrawal, please see our Privacy Policy. 

Reviewer #1: No 

Reviewer #2: No

---

## [Editor Report · Decision Letter 2]

31 Jan 2022

Direct-from-specimen microbial growth inhibition spectrums under antibiotic exposure and comparison to conventional antimicrobial susceptibility testing

PONE-D-21-08657R2

Dear Dr. Gau,

We’re pleased to inform you that your manuscript has been judged scientifically suitable for publication and will be formally accepted for publication once it meets all outstanding technical requirements.

Kind regards,

Mehmet A Orman

Academic Editor

PLOS ONE

---

## [Editor Report · Acceptance letter]

7 Feb 2022

PONE-D-21-08657R2 

Direct-from-specimen microbial growth inhibition spectrums under antibiotic exposure and comparison to conventional antimicrobial susceptibility testing 

Dear Dr. Gau:

I'm pleased to inform you that your manuscript has been deemed suitable for publication in PLOS ONE. Congratulations! Your manuscript is now with our production department. 

Kind regards, 

on behalf of

Dr. Mehmet A Orman 

Academic Editor

PLOS ONE